# Amino Acid-Related Metabolic Signature in Obese Children and Adolescents

**DOI:** 10.3390/nu14071454

**Published:** 2022-03-30

**Authors:** Nella Polidori, Eleonora Agata Grasso, Francesco Chiarelli, Cosimo Giannini

**Affiliations:** Department of Pediatrics, University of Chieti, Via dei Vestini 5, 66100 Chieti, Italy; nella.polidori@hotmail.it (N.P.); elegrasso601@gmail.com (E.A.G.); chiarelli@unich.it (F.C.)

**Keywords:** BCAAs, obesity, pediatrics, NAFLD, insulin resistance

## Abstract

The growing interest in metabolomics has spread to the search for suitable predictive biomarkers for complications related to the emerging issue of pediatric obesity and its related cardiovascular risk and metabolic alteration. Indeed, several studies have investigated the association between metabolic disorders and amino acids, in particular branched-chain amino acids (BCAAs). We have performed a revision of the literature to assess the role of BCAAs in children and adolescents’ metabolism, focusing on the molecular pathways involved. We searched on Pubmed/Medline, including articles published until February 2022. The results have shown that plasmatic levels of BCAAs are impaired already in obese children and adolescents. The relationship between BCAAs, obesity and the related metabolic disorders is explained on one side by the activation of the mTORC1 complex—that may promote insulin resistance—and on the other, by the accumulation of toxic metabolites, which may lead to mitochondrial dysfunction, stress kinase activation and damage of pancreatic cells. These compounds may help in the precocious identification of many complications of pediatric obesity. However, further studies are still needed to better assess if BCAAs may be used to screen these conditions and if any other metabolomic compound may be useful to achieve this goal.

## 1. Introduction

The epidemic of childhood and adolescent obesity still remains one of the most relevant issues worldwide and determines an increased risk of development of metabolic and cardiovascular diseases (CVD), also during childhood and adolescence, including insulin resistance (IR), dyslipidemia, altered glucose metabolism and type 2 diabetes mellitus (T2DM), arterial hypertension, hyperuricemia, non-alcoholic fatty liver disease (NAFLD), and others [1]. In recent years, advances in the use of the metabolomic approach as a tool for human disease research has determined an increased interest about the possibility of defining predictive biomarkers for cardiometabolic diseases and the pathophysiological mechanisms related to obesity complications [2,3,4]. Several studies investigated the association between IR or T2DM and amino acids (AAs) and their metabolites, in particular branched-chain amino acids (BCAAs), aromatic AAs and acylcarnitines [5,6,7,8]. AAs, known for their role as the building blocks of proteins, are also essential component signals implicated in direct and indirect metabolic processes as well as they are effectors or/and probably biomarkers for cardiometabolic diseases especially in the obese population. BCAAs (namely leucine, isoleucine and valine), are strictly implicated in the regulation of body weight, muscle protein synthesis and glucose homeostasis. In contrast with these health outcomes, reports have underlined that high levels of BCAAs are strongly associated with an increased risk of IR and T2DM, together with glucose, insulin and inflammatory markers. Moreover, to date the key role of AAs in obesity or obesity-related diseases remains unknown. In fact, several studies have discussed whether BCAAs are a causative factor in IR and T2DM or biomarkers of altered insulin signaling. Further studies are needed to better clarify these mechanisms. This crucial role of AAs was suggested 40 years earlier by Felig et al., who showed, for the first time, increased levels of BCAAs and aromatic AA (phenylalanine and tyrosine) and decreased levels of glycine in obese compared to normal-weight subjects. These elevated levels of BCAAs were associated with IR status. Afterwards, dyslipidemia become a focus, supported by the evidence that lipids could function as mechanistic drivers of IR and T2DM [9]. Subsequently, Newgard et al. demonstrated a strong association between BCAAs, aromatic AAs and acylcarnitines with IR expressed by the homeostasis model assessment IR index (HOMA-IR) [5]. In addition, increased levels of circulating BCAA at baseline have been documented in adults as predictive of future onset of T2DM during 7 to 12 years of follow-up [10,11]. Moreover, other AAs change related to health status. For example, serine and glycine levels are decreased while glutamate levels are increased in subjects with obesity, IR and NAFLD [12,13]. Most of these studies have been conducted on adult population, but the increased rate of obesity particularly in childhood has suggested the need to expand the knowledge about the bases and consequences of these diseases in youth in whom a relatively shorter duration of obesity, ongoing linear growth and pubertal hormones should be considered. In agreement with adults’ studies, the metabolic profile of a large Hispanic cohort of children showed altered BCAAs, IR, mitochondrial disorder and dysfunction of fatty acid beta oxidation [14]. Other studies in pediatric populations have documented analogous results [8,15,16,17,18]. In opposition, in a study conducted on obese adolescents with and without T2DM, results did not demonstrate impaired fatty acid and amino acid metabolism [19].

Moreover, Goffredo et al. [20], in a cohort of obese adolescents with or without NAFLD assessed by MRI, documented higher plasma levels of BCAA (valine, isoleucine and lysin) in adolescents with NAFLD compared to adolescents without NAFLD. In addition, BCAAs were negatively correlated with insulin sensitivity. Therefore, adolescents with NAFLD showed a dysregulation of AAs profile independent of obesity and IR, predicting an increase in hepatic fat content over time. The pathophysiological pathway of high levels of BCAA in obesity is not completely understood but could implicate chronic low-grade inflammation by prompting pro-inflammatory gene expression in adipose tissue, determining further obesity effects on metabolic health and an increased risk of cardiovascular and liver disease [21,22,23]. This finding may open interesting possibilities for early-start prevention and treatment strategies on high-risk subjects in order to prevent further development of T2DM, liver disease and CVD. In addition, AAs evaluation is very easy, so, it may increase patients’ compliance compared to MRI for quantifying the accumulation of liver fat [24]. However, it must be taken into account that nowadays the analysis required to assess the AAs levels is expensive in some countries and not part of the normal services routinely provided in the clinical setting. However, it may represent a promising revolution approaching in the near future.

In this review we aimed to mainly highlight the association between AAs and obesity, IR and metabolic syndrome (MetS) and its related effects on glucose and liver metabolism in adolescents.

## 2. Epidemiology of Pediatric Obesity and Its Major Risk Factors

Childhood and adolescent obesity have reached epidemic proportions worldwide. In fact, the Global Burden of Disease Study has shown that the prevalence of obesity in children has increased of two-folded in more than 70 countries since 1980, reaching a global prevalence of 5% or 23% in 2015, respectively, in obesity or overall overweight and obesity [25,26]. Although in recent years the prevalence of obesity in younger children reached a plateau or smooth decrease, an increase afterwards has been documented with many youths showing a condition of extreme obesity [27,28]. Moreover, the number of adolescents with obesity is yet increasing [29]. The prevalence of obesity changes with ethnicity and socioeconomic factors. For example, in the US one-third of children and adolescents are overweight or obese. In particular, 22.8% of preschool children (aged 2–5 years), 34.2% of school-aged children (aged 6–11 years), and 34.5% of adolescents (aged 12–19 years) are affected by overweight or obesity, and 8.4% of preschool children (aged 2–5 years), 17.7% of school-aged children (aged 6–11 years), and 20.5% of adolescents (aged 12–19 years) are obese [30]. Pediatric obesity is a multifactorial condition characterized by a complex interaction between genetic and non-genetic factors [31]. In most cases, it is caused by an imbalance between excess energy intake and lower expenditure, also influenced by genetics and social factors, as well as ethnicity, socio-economic status, environmental factors and the body’s predisposition to obesity established by genetics or epigenetic programming [32].

Various genes have been identified related to obesity, such as mutation in melanocortin 4 receptor gene, the single nucleotide polymorphism on the fat mass and obesity-associated gene, defects in leptin, leptin receptor, proopiomelanocortin, and proprotein convertase [30,33], but although genetic defects increase individual susceptibility to obesity, they explain for less than 1% of the cases [34]. So, genetic susceptibility plays an important role when coupled with other contributing environmental and behavioral factors. Environment factors prompting children’s food intake and physical activity are extremely complex, including family habits on food type and amount, mealtime, dining out, and lifestyle. Lack of physical activity and increased sedentary time also play important roles in the development of obesity. Moreover, numerous reports have suggested that unhealthy weight gain frequently occurred during the lockdown imposed by the COVID-19 pandemic [35,36,37]. Several authors have speculated that, compared with the pre-pandemic period, both BMI-z-scores, and childhood obesity prevalence under COVID-19 would rise, and the magnitude of the increase would be proportional to the length and severity of the pandemic [38,39]. Recently, Vogel et al. compared the trends of BMI changes and the proportions of high positive (HPC)/negative weight changes (HNC) from 2005 to 2019 with the respective changes from 2019 (pre-pandemic) to 2020 (after the onset of anti-pandemic measures) in a large pediatric cohort in Germany. They found a substantial weight gain across all weight and age groups, reflected by an increase in the 3-month change in BMI-SDS, an increase in the proportion of children showing HPC, and a decrease in the proportion of children showing HNC. Thus, in this recent study reported data have significantly supported the dramatic effects of pandemic on the still-growing problem of childhood obesity. In fact, these effects increased by more than 30 times within a relatively short period, suggesting that changes in health-related behavior lead to a significant further aggravation of the childhood obesity pandemic [39]. Moreover, Shlomit et al., in a retrospective cohort study, analyzed changes in BMI among a large Israel population of children, adolescents and young adults during the COVID-19 pandemic in Israel. The COVID-19 pandemic correlated with general weight gain among children and adolescents. In particular, for 21,610 individuals (35.6%), BMI-SDS increased by ≥0.25 SD. The increase in BMI-SDS was greater in children aged 2–6 years. In addition, they smartly demonstrated that overweight and obesity presented in 11.2% of those with normal weight in the pre-pandemic period and obesity presented in 21.4% of those with overweight in the pre-pandemic period [40]. Taken together, these results and further confirming data reported in the very near future will strongly highlight the relevant and global burden of childhood obesity and its related parallel increase of metabolic complications.

Notably, the rise in the incidence of obesity would be mirrored by an increasing number of cases of early alteration of glucose metabolism, prediabetes and youth-onset T2DM related to obesity [41]. As is known, obesity has been defined as the major cardiometabolic risk factor, including various co-morbidities such as IR, impaired glucose tolerance, T2DM, hypertension, dyslipidemia and NALFD [42]. These injuries, strongly linked to obesity, are components of MetS, defined also in the pediatric population [43]. MetS is not considered a set of various metabolic comorbidities strongly related to the adipose tissue in the body, but it represents a prominent risk factor for the progression towards CVD.

Recent data have shown that obesity was responsible for about four million deaths worldwide in 2015, and about 70% of these were strictly related to obese-related developed CVD [27]. Not only metabolic comorbidities, but also an increased risk for malignancies, musculoskeletal, pulmonary, gastrointestinal and neurologic disorders have been associated to excess body weight. The intricacy of risk factors for rising obesity between children and adolescents determines the difficulty in treatment for the pediatric population. Many interventional trials for childhood and adolescent obesity have shown only very limited success to date. Therefore, early identification and prevention is fundament to prevent the global epidemic of obesity and its complications.

## 3. Main Complications Associated with Pediatric Obesity

Childhood obesity can affect every system of the human body, with consequences that may manifest in different ways and at different ages. Although some obese patients do not experience any conditions [44], the severity of the comorbidities and complications and particularly IR, impaired blood glucose metabolism, blood pressure (BP) alterations, liver and kidney damages, usually correlates with BMI [45]. More importantly, several studies have shown that being obese as a child increases the risk of being obese during adulthood [46]. The cooccurrence of these complex metabolic alterations related to obesity in childhood defined the risk of developing a severe metabolic status, commonly defined as MetS. This condition is defined as a spectrum of conditions that share the same pathophysiology [47], and are known to increase the risk of CVD, T2DM and all-cause mortality [44]. The main features include obesity, IR, atherogenic dyslipidemia and hypertension, but in some not-yet-universally accepted definitions, it is also related to other IR-related conditions like polycystic ovary syndrome (PCOS) and NAFLD [44]. The cause and the pathogenesis behind it are not completely understood; however, a large number of studies have clearly shown that IR has a central role in the development of MetS and its complications [48]. Obesity in childhood is also correlated to an increased risk of developing impaired BP. Hypertension in children is defined by systolic or diastolic BP above or equal to the 95th percentile for age, sex and height in repeated measurements. Body weight correlates with high values of BP and the risk for organ damage is directly related to the duration of the condition [49]. Among obese children and adolescents, the high prevalence of IR correlates with impaired 24 h ambulatory BP monitoring and T2DM [45,50]. In addition, T2DM rate increases proportionally with the prevalence of pediatric obesity, which is concerning considering the high chances of undergoing microvascular and macrovascular damage in early adulthood [44]. Obesity and IR also exert deleterious effects on the liver, where fat deposition leads to NAFLD. This effect results in a wide spectrum of liver conditions, ranging from asymptomatic steatosis to steatohepatitis. The risk of progression seems to be mainly related to the obesity degree and the presence of NAFLD strongly modulates longitudinally 2 h blood glucose, biomarkers of IR, and hepatocellular apoptosis in obese youth [51]. The rising prevalence of NAFLD in children is alarming since it can progress to cirrhosis and end-stage liver disease. Thus, recent data have clearly shown that liver steatosis represent one of the most important emerging liver diseases in developed countries [52] and represents an important risk factor for the development of T2DM and a component of the MetS [53]. Liver steatosis is expected to become one of the most common causes of hepatic disease in children and young adults [54], paralleling the increasing prevalence of childhood obesity. Cohort studies have shown a prevalence of ∼50% of liver steatosis in the pediatric obese population with a significant male predominance [55,56] with a high prevalence also in the prepubertal age ranging from around 30% to 52% [57,58], and significantly increases with age reaching significantly higher values during puberty [55,56].

However, the reason for male preponderance is not completely understood. Current evidence suggests that the differences in sex hormones may play a role, on one side, by estrogens’ immunomodulant role that may interfere with inflammation in the liver. On the other side, by the modulation of body fat distribution, as demonstrated by the shift to the abdominal position, likely to happen after menopause [58].

Obesity and IR may predispose girls to the onset of PCOS, an endocrine disorder characterized by ovulatory dysfunction (expressed as oligo or anovulation), hyperandrogenism and the appearance of polycystic ovaries on ultrasound [59]. Although the relationship with IR and consequently MS is evident, the contribution of BMI to PCOS is still the object of debate [60]. In addition, increased adiposity depots represent a relevant risk factor inducing the development of impaired respiratory function. Particularly, obstructive sleep apnea syndrome (OSAS) is a respiratory condition characterized by upper airway obstruction with the alteration of gas exchange during sleep; patients usually snore at night and complain of daytime sleepiness, academic difficulties, enuresis and hypertension [60]. The obstruction is due to many factors, including adenotonsillar hypertrophy, craniofacial abnormalities and rhinitis. Obesity takes part in the pathogenesis of OSAS, either by reducing the caliber of the airways’ lumen because of the presence of fat in proximity to the pharyngeal soft tissues or by reducing respiratory function because of the higher adipose content in the thorax and the abdomen [60]. Thus, the management of these patients should always incentivize weight loss.

High BMI can also affect bone development and functionality, increasing susceptibility to fractures and damaging the epiphyseal growth plates, resulting in pain and limited mobility [59,61]. Besides the systems previously noted, other systems can be affected by obesity: children can develop acanthosis nigricans due to IR, sequalae to skin friction (e.g., intertriginous irritation or infections) and stretch marks. Moreover, among the neurological disorders, pseudotumor cerebri is most commonly reported in obese post-puberal girls; herby, the most effective treatment is known to be weight loss [62]. Obesity has been identified as a possible risk factor for the onset of multiple sclerosis in children and young women [63]; interestingly, the number of relapses and response to treatment is correlated to BMI [64]. Finally, psychological and psychiatric factors (depression, anxiety, eating disorder, stress, body shape concerns, low self-esteem) are all common comorbidities that have a huge impact on quality of life for these children.

## 4. Early Markers of Obesity-Related Complications

Microvascular damage and obesity-related dyslipidemia can appear in early life and are correlated with high BMI during childhood [43]. Therefore, it is paramount to promote a precocious identification of the complications related to the disorder with the aim to activate a prompt intervention for reducing the risk of developing the disease. Several both anthropometric and metabolic indexes have been shown to be useful. Particularly, this was part of the rational of the identification and prevention of dietary- and lifestyle-induced health effects in children and infants (IDEFICS) study [42], which aimed to promote MS prevention in obese children. The IDEFICS study [42] identified a set of parameters in order to detect components of MS in children aged 2–11 years. These included obesity assessed by waist circumference and other metabolic data namely triglycerides, HDL cholesterol, BP, fasting glucose or insulin. According to authors, if three or more of these parameters exceed the 90th percentile (or ≤10th percentile for HDL cholesterol) monitoring should be started, while intervention should be started when three or more exceed the 95th percentile. Therefore, clinical evaluation and follow-up of a child overweight or obese requires weight, height and BMI calculation and monitoring on sex- and age-appropriate growth charts. This should be performed at least annually during well-child and sick-child visits [47]. To more accurately measure the quantity of adipose tissue, skinfold thickness, waist-to-height ratio and waist circumference should be measured, as well as a bioelectrical impedance analysis. Despite the reduced reliability of anthropometric markers in early childhood, a diagnosis of obesity can be made in children under 2 years of age if the sex-specific weight for recumbent length is over the 97.7th percentile on the WHO charts [65]. However, Chiarelli et al. encourage monitoring younger children and toddlers that were at increased risk [47].

Performing a 2 h oral glucose tolerance was found to be the most effective way to assess insulin sensitivity and secretion in children [35,66]; HbA1c may be useful to diagnose diabetes for values above 6.5% on two measurements, even though this tool is less reliable on children than adults [67]. Several markers have been studied with the aim of identifying early atherosclerosis. The C-reactive protein was reported to be elevated in obese children, supporting the low-inflammatory state which may be the set of endothelial vascular damage [68,69]. However, a lack of significance was found with early onset CVD [70], scaling down its role as a biomarker. Several studies reported abnormalities in the lipid panel, with a positive association between BMI, IR, myocardial infarction, NAFLD and increased VLDL and IDL values, whereas HDL particle size is known to be protective [71,72,73]. Moreover, a triglyceride/HDL-cholesterol ratio greater than or equal to 3.5 was proposed as a marker of cardiovascular risk in obese children [74]. In order to monitor or exclude fat deposition in the liver, cut-offs for alanine aminotransferase (ALT) were revised, identifying values below 25 U/L for boys and 23 U/L for girls [75]. Indeed, fatty liver is frequently present despite normal levels of ALT [67]; thus, ALT may be suggestive of a more advance stage of the disease (NAFLD or fibrosis) [76]. In order to early diagnose other complications, other exams (e.g., hormonal profiles to screen PCOS) should be requested in case of the occurrence of clinical symptoms that may suggest any other disorder. Although these markers are widely reported to properly characterize the obesity-related complications, most of them are affected by several limits. Particularly, most of them might be documented only after the onset of the complications. In addition, there is a growing interest in the study of metabolic components such as adipocytokines, gut peptides, cardiovascular markers, and inflammatory marker; thus, if these are altered among children affected by obesity and their complications, dosing these substances may represent an effective tool to screen the onset of complications. Among these metabolites, AAs and acyl carnitines have been proposed as suitable markers to predict complications in the pediatric population [19].

## 5. BCAAs: Function and Signaling

BCAAs leucine, isoleucine and valine belong to the 9 essential AAs in the human diet (Figure 1) [77].

BCAAs can be stored in different parts of the cell and directed towards either anabolic processes (e.g., sterols biosynthesis or ketogenesis) or shuttled into the mitochondria for oxidation. The metabolism of BCAA stakes several steps and starts primarily in skeletal muscle. The first step consists of deamination, which begins by cutting the amino group from the carbon backbone and transferring it to the α keto-acidin, a process catalyzed by the mitochondrial branched-chain amino transferase (BCATm). The final step is catalyzed by the multienzyme complex BCKA dehydrogenase (BCKDH) in the mitochondria, with formation of branched-chain acyl-coenzyme A esters of the BCKA precursors [77].

Behind their typical nutritional and anabolic function, BCAAs can function as signaling molecules and take part in several metabolic processes, such as oxidative stress regulation or hormone release, thus finally influencing glucose metabolism [77,78,79]. Leucine and isoleucine are insulin secretagogues, therefore, chronic elevation of these AAs may contribute to hyperinsulinism initially and beta-cell failure later [17,80]. Lynch et al. suggested two mechanisms that may explain the relationship between BCAAs and IR/T2DM [79]. According to the first one, an impaired metabolism of BCAAs may lead to an increased number of toxic compounds in the mitochondria, causing toxic damage in pancreatic beta cells and consequently insulin secretion impairment. This interesting theory is supported by studies on metabolic AAs disorders (e.g., maple syrup disease or orotic aciduria). The second one comes from the acknowledgment of the additive effect of BCAAs and insulin on cellular processes that involve the mTORC1 pathway. In fact, studies have shown an effect of BCAAs excess on the activation of mTORC1, thus leading to IR [5,81]. Therefore, the complete knowledge of the relationship between BCAAs pathways and mTORC signaling are paramount to offer novel insights in obesity-related complications in children.

### BCAA and mTORC Signaling Pathway

BCAA enhances protein synthesis up-regulating mRNA transcription in the skeletal muscle, with a permissive effect triggered by insulin [82]. In respect to that, leucine has a stimulatory effect on several targets: (a) the eIF4F complex, which plays an important role at the start of mRNA translation, (b) the phosphorylation status of eIF4G, S6K and S6, which are all substrates of the mammalian target of rapamycin (mTOR) [82]. mTOR is a highly conserved serine–threonine protein kinase that regulates cell growth in response to nutrient status, acting as a sensor for the intracellular availability of AAs. mTOR is the catalytic subunit of the two protein complexes mTORC1 and mTORC2 and plays a central role in the regulation of cell growth, motility and metabolism in response to a wide range of environmental triggers [83]. mTORC1 modulates protein synthesis and cell growth in response to nutrients through its downstream effectors (e.g., 4E-BP1 or EIF4E-BP1, S6K); meanwhile, mTORC2 regulates cytoskeletal remodeling and cells’ proliferation, migration and survival via PI3K and growth factor signaling [83]. Therefore, mTORC1 promotes anabolic processes (e.g., protein, lipid and nucleotide synthesis) in response to nutrient-rich condition, primarily AAs and growth factors. On the other hand, when the cell is facing stressful conditions (such as amino acid deprivation), its inactivation leads to autophagy [84]. The mechanism of mTORC1 activation by AAs is not completely understood; a key role might be played by Rag GTPase, which binds mTORC1 in an amino acid-dependent way and promote its translocation in lysosome where the complex interacts with Rheb-GTP, causing mTORC1 activation [85].

More evidence about the link among mTORC and AAs and, in particular, BCAAs, comes from animal studies. A group of authors described the effect of the inhibitor of mTORC1 rapamycin, on the BCATm knock-out mouse. In detail, the KO-model displayed elevated concentration of BCAAs because of its deficient metabolism and, concurrently, it developed organ hypertrophy. This was related to an abnormal mTORC1 activation, as supported by the increased levels of its substrates (S6, 4E-BP1). Moreover, rapamycin was able to prevent organ enlargement supporting the central role of mTORC1 in BCAAs-mediate protein synthesis [86,87,88]. In another study conducted on rats, rates of protein synthesis were compared in response to stimulation with insulin, AAs and then infusing both [85]. In the first case it was reached the maximal stimulation of Akt, proving the key role of PI3K/Akt for insulin-induced mTORC1 activation, although there was no effect on protein synthesis. On the contrary, infusing AAs without insulin lead to moderate protein synthesis and the activation of mTORC1. Finally, infusing a combination of both produced mTORC1 stimulation with the maximum rate of protein synthesis. Indeed, since both AAs and insulin were reported to enhance the activation of mTORC1, many studies have been conducted to assess the mechanism of these processes in order to understand if BCAAs may be related to the onset of IR. In rodent models of obesity, an increased activation of mTOR was reported, suggesting again that a persistent activation of mTORC1 and S6K may cause elevated IRS1 phosphorylation, consequently promoting IR [89]. However, many questions revolve around the relationship between BCAAs, mTORC1 and IR. Thus, the same pathway is activated by exercise, a known protective factor for IR, and BCAAs intake was proved to have a beneficial effect on human metabolism [90,91]. Moreover, it is not certain that IRS1 and IRS2 are sensible enough to these small changes on BCAs level to explain the link with IR. BCAAs may also influence protein degradation inhibiting the ubiquitin–proteasomal pathway and autophagy, similarly mediated by the mTORC1 and Akt cascade [79]. Indeed, AAs inhibit autophagy in rat hepatocytes, in a process that is mediated by the phosphorylation of the ribosomal protein S6. It is also known that leucine, together with tyrosine and phenylalanine, is able to stimulate S6 kinase inducing S6 phosphorylation [92]. Therefore, activation of mTORC pathways are modulated by amino acid, insulin, glucose and growth factors thus affecting several metabolism (Figure 2). Thus, further studies exploring the link between of BCAAs, mTORC1 and IR are needed to fully understand the complex mechanisms involved in obese related complications.

## 6. The Association of AAs in Obesity, IR and Metabolism in Youth

High-protein diets are universally considered an effective way to promote weight loss, increase satiety and maintain lean mass [79], with studies proving their beneficial effects for metabolic disorders [93]. On the contrary, a link between diets rich in red meat, animal protein and specific amino acid groups with general body obesity has also been proven [94,95]. These factors may also increase the risk of T2DM [96,97,98]. An association between high BCAAs levels, CVD and metabolic disorders such as IR and T2DM has been proposed [9,17,20,79,99,100,101,102,103,104], with contrasting results reported in literature, likely due to several factors, such as the constituents of the regimen administered and the presence of comorbidities [99].

As previously mentioned, two principal mechanisms have been proposed regarding the link between BCAAs and metabolic diseases (Figure 3). The first one involves the persistent activation of the mTORC1 complex, whereas the second one revolves around BCAAs’ metabolism impairment, which would lead to BCAAs’ metabolites accumulation initially, and mitochondrial dysfunction, stress kinase activation and cell apoptosis later [79]. All these phenomena are associated with metabolic disorders, such as IR and T2DM, though further studies are required to better assess the pathophysiology behind them.

Most of the knowledge on the matter comes from metabolomic studies conducted in animal models or in adult cohorts [5,79,105,106,107] in which high levels of BCAAs were detected in presence of IR and/or increased levels of HbA1c, suggesting that they may be biomarkers that predict the onset of T2DM [10,11,108]. However, considering the huge impact of pediatric obesity on global health, there is a growing interest in researching biomarkers in younger populations.

A recent study was conducted to assess the metabolomic profiles of a large cohort (*n* = 523) of adolescents, classified in four different phenotypes (non/overweight or obese matching with low or high metabolic risk) [109]. Among the metabolites analyzed, BCAAs were higher in the obese/overweight adolescents’ group with high metabolic risk. Similarly, Short et al. detected a 10 to 16% increase of all the BCAAs in the group of obese adolescents (*n* = 58) compared to the control group: higher levels of aromatic AAs (phenylalanine and tyrosine), glutamate, lysine and its metabolite 2-aminoadipic acid were found in these patients, whereas glutamine, GABA and valine metabolites were lower [78]. A rise of BCAAs levels was also reported in a Hispanic population of obese children; however, levels of other compounds, such as BCAAs catabolites (2-methylbutyrylcarnitine, 3-methyl-2-oxobutyrate, and isovalerylcarnitine), propionylcarnitine and butyrylcarnitine, ketoacids as a-hydroxybutyrate and a-ketobutyrate (AKB) were elevated [14]. Moreover, other AAs (e.g., tyrosine, alanine, phenylalanine, glutamine and lysine) were increased, whereas levels of asparagine, aspartate, glycine, serine and histidine were lower in obese children. This different metabolomic profile found was consistent with increased levels of inflammation, oxidative stress and IR [14]. IR has been correlated with other metabolic disturbances such as reduced substrates influx in the Krebs cycle [110], reduced lipolysis and fatty acid oxidation; consequently, a higher mitochondrial activity of pyruvate oxidation leads to increased levels of alanine and lactate [111]. Indeed, increased levels of BCAAs have been related to higher levels of the HOMA-IR index in several studies [5,112,113]. Likewise, a cohort of 109 Korean boys from the Korean Child Obesity Cohort Study was screened in order to assess specific biomarkers predictive of MS and IR [112]. In this study, 186 metabolites were analyzed, and, among these, higher levels of BCAAs, aromatic AAs and acylcarnitines were reported. Since C3 and C5 carnitines are products of BCAA’s metabolism, the rise of the first ones could be the product of the higher levels of BCAAs. Moreover, baseline isoleucine and valine were positively associated with future HOMA-IR and metabolic risk score. Levels of BCAAs and their metabolites were significantly high in a cohort of 82 obese adolescents during fasting state, with a significant difference in boys rather than girls [8]. A sex-dependent difference was also found for IR-related biomarkers (such as adiponectin, triglycerides and HDL) and HOMA index. Similar results were reported by a Mexican transversal study which included 608 Mexican young adults (mean age 19.9 ± 2.4) divided in two subsets in relation to BMI and/or IR. In detail, results showed that the control group displayed lower levels of plasmatic AAs (alanine, arginine, aspartic acid, cysteine, isoleucine, leucine, phenylalanine, proline, taurine, tyrosine and valine levels) and glycine levels [114]. Furthermore, the amino acid profile showed a different pattern according to a HOMA index < 2.5, even though both conditions shared a higher plasmatic level of alanine, aspartate, proline and tyrosine and a lower level of glycine.

The relationship between these compounds and cardiometabolic abnormalities was also supported by a case–control study conducted on individuals affected by genetically inherited disorders of BCAAs metabolism, such as propionic acidemia [20]. This group demonstrated how patients affected by organic acidemias developed fasting hyperglycemia, dyslipidemia, abdominal adiposity and IR aside from dietary adjustments. These conditions were thought to be the result of the generation of reactive oxygen species and the ectopic lipid storage as a consequence of the abnormal mitochondrial function.

In order to assess BCAAs behavior in relation to fasting and feeding state, a study group from Yale measured the plasma concentration of BCAAs, alfa and beta-hydroxy-butyrate and lactate in response to an oral glucose challenge in 78 non-diabetic adolescents [15]. The cohort was clustered in 3 groups according to the whole-body sensitivity index. The study found early metabolic perturbation in subjects with impaired insulin sensitivity. Higher levels of BCAAs and alfa-hydroxy-butyrate were found in the same group during the course of the OGTT. Moreover, increased levels of alfa-hydroxy-butyrate were associated with worsening of the IR over time, suggesting that this compound may be useful to predict the future outcome of these children. IR was inversely correlated with BCAAs levels in children with normal weight, giving credit to the importance of obesity in setting the metabolic abnormalities that might interfere with AAs metabolism, and consequently, to their elevation in the serum. Finally, a group of authors questioned the relationship between BCAAs and obesity in adolescents dosing these metabolites in the urine [111]. The cohort was divided in three groups, including 30 obese patients with T2DM, 30 obese patients without other conditions and 30 healthy controls. BCAAs and their metabolites were the most represented metabolite among the T2DM group compared to the obese the healthy adolescents. Although the urine metabolites reached higher levels in the obese group than among controls, this difference was not judged significant. Interestingly, comparing these results with the ones found in plasma, no significant difference of BCAAs concentration was found among the T2DM and the obese group, suggesting that urine BCAAs may represent a more specific marker for diabetes rather than in plasma, were they correlate with obesity and IR independently from T2DM [115].

Other AAs have been proposed as possible markers of obesity and cardiometabolic unhealthiness [113]. Among these, higher levels of phenylalanine and tyrosine were found in children affected by obesity [14,16,17,71]. An alteration on tyrosine metabolism could precede the BCAAs’ rise, because both compete for the same cellular transporter [116]. Interestingly, in other studies, tyrosine was the only metabolite significantly associated with obesity and IR [14,116]. This may be explained either by the stimulating effect of tyrosine on insulin secretion and vice versa or by insulin action on tyrosine metabolism. However, several studies reported opposite results regarding the relationship between BCAAs and obesity. Following the reported discovery of the impact of elevated AAs on the onset of T2DM in adults, Michaliszyn et al., investigated beta-cell functionality in a cohort of young patients [117]. The results of this study conflicted with what was previously reported, since the increased plasmatic amino acid concentration was positively associated with beta-cell functionality and, consequently, to a lower risk of T2DM. However, the authors did not exclude a time-dependent failure of beta-cells that might explain this variance among the different age groups. Similar results were found in a prospective study of 253 children (age ranged from 6 to 10 years), in which BCAAs were not associated with a worsening metabolic status, which was associated with a reduction in glucose levels during fasting among boys [118]. Also, Hellmuth et al. reported normal levels of BCAAs in his cohort; however, the alteration of BCAAs metabolite ratio may be suggestive of a prior reduction of BCAAs metabolism that could precede their rise in obese and IR patients [116].

### BCAAs as a Marker of Dietary Composition

In order to better explain the evidence suggesting the association between the alteration of metabolic profiles and the development of CVD, several studies have tried to explore the metabolic variations in response to dietary habits [119]. Of note, most of these available data have concluded that these profiles change across different diets. The related metabolites originate directly from food, beverage and additives, or may be produced by gut microbiota. Among the compounds explored, BCAAs’ levels were found to change in response to the dietary regimen followed. For example, the Dietary Approaches to Stop Hypertension (DASH) diet is a dietary pattern rich in fruit, vegetables, low-fat dairy products and low in sugar-sweetened beverages, sweets and red meat. In this study, levels of β-hydroxyisovalerate—a metabolite of BCAAs—were lower in the individuals who followed the DASH diet, compared to the ones randomly assigned to the control diet [120]. Others searched for biomarkers variations in response to the Mediterranean diet, finding once again that BCAAs level decreased following a healthy dietary regimen, suggesting that these compounds may be used as a marker of effectiveness of an unhealthy diet [121]. Therefore, metabolomic profiling may be used to assess the quality of nutrition and patients’ compliance to the diet, but also to help clarify the mechanism that drive the health benefits of specific nutrition [122]. However, further longitudinal studies are need in order to properly characterize the direct effect of diet regimes on BCAAs’ levels with a direct cause effect on body metabolism.

## 7. AAs Related Effect on Glucose and Liver Metabolism

Several biomarkers have been associated with the presence of NAFLD and to the future risk to develop the disease [73,123]. Recent studies in the adult population have consistently reported increased concentrations of BCAAs and their metabolites in patients affected by NAFLD and non-alcoholic steatohepatitis [12,124,125,126,127,128,129], correlating BCAAs with hepatic fat storage [73,129]. Less is known about NAFLD in the pediatric population, with few metabolomic studies published so far. The development of a screening panel has been proposed by an American group which performed liquid chromatography–mass spectroscopy on 559 plasma samples from a cohort of patients aged from 2 to 25 years old, 222 of which with NAFLD [130]. Among the compounds identified in the NAFLD group, five were adducts to the amino acid serine, leucine/isoleucine and tryptophan.

A study conducted on 78 obese adolescents with or without NAFLD proved the relationship among BCAAs’ plasmatic concentration and intra-hepatic fat content independently from the presence of IR [20]. Moreover, higher baseline valine levels were predictive of fat accumulation during the two-year follow-up (*p* = 0.01), suggesting that an early alteration in BCAAs’ metabolism might affect the risk of NAFLD developing and/or worsening among obese pediatric patients [20]. In the same year, a prospective study published the results of a 7-year and 10-year follow up of the population-based Young Finns study [73]. Liver fat content was assessed in 2002 patients, revealing increased levels of sixty compounds from different metabolic pathways in the patients with fatty liver (*n* = 372). Among these, circulating BCAAs were found elevated in the patients affected and correlated with the future risk of developing NAFLD. Authors suggested that these alterations may be the consequence both of mitochondrial dysfunction and to the presence of IR. However, such a variety of abnormal metabolites levels from different pathways (e.g., glycolysis and gluconeogenesis) could related to a perturbation of systemic metabolism that may precede the onset of NAFLD. Indeed, alterations in the metabolomic profile of these patients were present 10 years’ prior the diagnosis of fatty liver, suggesting how these compounds may be a suitable biomarker for the disease [73].

More recently, Lyscka et al., assessed the amino acidic profile, fat liver content and metabolic parameters (e.g., HOMA-indices, liver transaminases, gamma-glutamyl transferase, total cholesterol, insulin, triglycerides, ferritin, procalcitonin, TNFα and CK-18) of 68 children in order to investigate the role of BCAAs in high-risk children and adolescence [24]. High levels of BCAAs have been determined in the cohort. Moreover, authors developed a model based on BCAAs’ levels able to predict liver fat content in these patients, suggesting an alternative to MRI and biopsy. Alterations on other major AAs have been correlated with NAFLD. Among these, tyrosine was reported to be dysregulated among obese adolescents, with levels that were proportional to the severity of NAFLD [131]. Indeed, since tyrosine is able to access the ketogenic pathway, in the set of an excessive caloric state, a high dietary intake of this amino acid might stimulate fatty acid synthesis and deposition in the hepatic tissue. Finally, studies conducted on patients who underwent bariatric surgery support the role of obesity in BCAA metabolism [132,133], since a reduction of these compounds was reported after the weight loss [132]. In a cohort of 6 adolescents (mean age 14.5 years) that underwent laparoscopic sleeve gastrectomy with a drop of the mean BMI from 48.4 to 37.1, a significant change in the metronome profile was detected at 6 months from the surgery [134]. The greater decrease was related to phenylalanine levels, whereas methionine was significantly increased. Interestingly, authors correlated the increase of phenylalanine to serum alanine transaminase stressing that this amino acid may be relevant as a biomarker of liver dysfunction associated with fatty liver disease [135].

Another interesting marker could be represented by the glutamate–serine–glycine (GSG) index, calculated dosing plasmatic amino acid and dividing glutamate by the sum of serine and glycine. Indeed, this index was found to be higher in adolescents with NAFLD compared to controls [136]. These compounds are implicated in glutathione synthesis, which is stimulated in response of the oxidative stress present in NALFD’s inflammatory environment, with an increase of glutamate and a reduction of serine and glycine. According to Leonetti et al., the GSG index was also associated with an increase of transaminase but was not related to hepatic IR and BCAAs levels. Indeed, the authors suggested that these two markers may be involved in NAFLD in different ways: BCAAs as a marker of IR and GSG index as one of oxidative stress.

## 8. Conclusions

Several studies found a correlation between BCAAs, obesity and metabolic affections such as IR, dyslipidemia and NAFLD. This link may be a useful tool in the early prediction of the onset of these complications in children and adolescents at higher risk.

The connection between BCAAs and IR is not completely understood; however, BCAAs may directly activate the substrates of the mTORC1 complex, which is known to be involved in different metabolic processes, such as cell growth or autophagy. Indeed, BCAAs’ metabolites may exert a toxic effect on pancreatic beta cells, which may lead to mitochondrial dysfunction, stress kinase activation and cell apoptosis initially and to IR later. Higher plasmatic levels of BCAAs have also been identified in patients with a high content of fat in their livers and in those diagnosed with NAFLD.

The development of a metabolomic screening panel for children and adolescents with obesity may help identify the patients that should undergo further testing and a more critical follow-up. Further studies are still needed to better assess if BCAAs should be added to this panel and/or which other compounds may be useful to achieve this goal.

## Figures and Tables

**Figure 1 nutrients-14-01454-f001:**
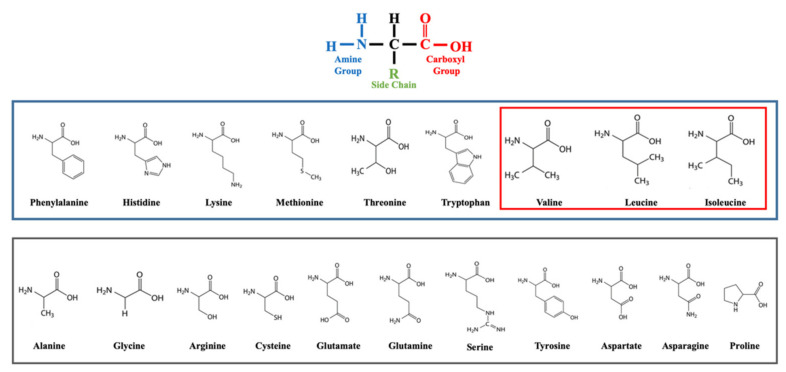
The 20 amino acids (AAs). At the top the basal structure of AAs. In the blue box, the nine essential AAs. In the red box, the three branched-chain AAs. In the grey box, the non-essential AAs.

**Figure 2 nutrients-14-01454-f002:**
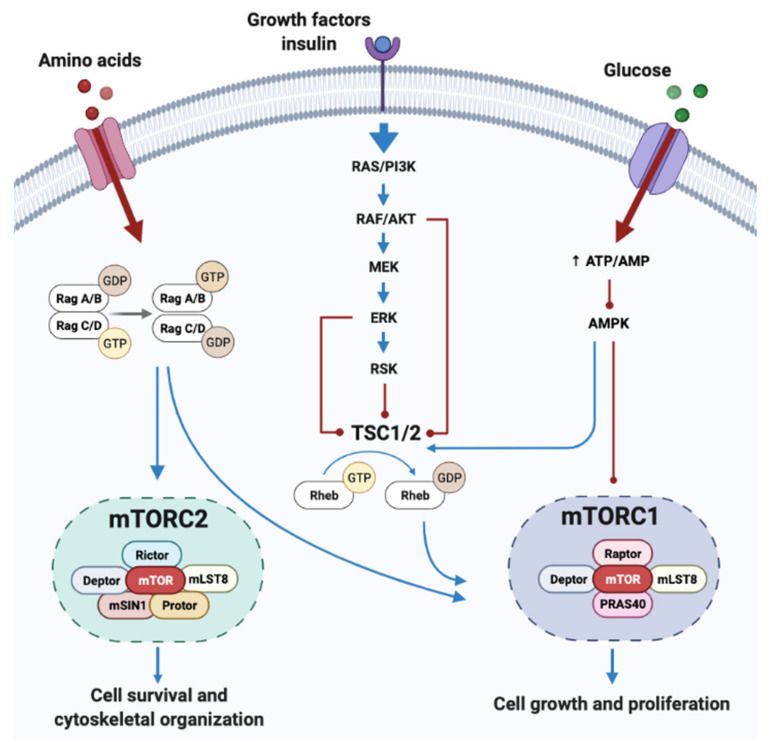
Main metabolic and catabolic effects of mTORC pathway activation. From the left to the right: amino acids activate the Rag kinases, which enhance cell survival and cytoskeletal organization toward mTORC2 complex activation. Growth factors and insulin triggers the RAS/PI3K cascade which leads to the activation of TSC1/2, thus allowing mTORC1 activation throw Rheb kinase. mTORC1 can be activated also by glucose via AMPK, and it ultimately leads to cell growth and proliferation.

**Figure 3 nutrients-14-01454-f003:**
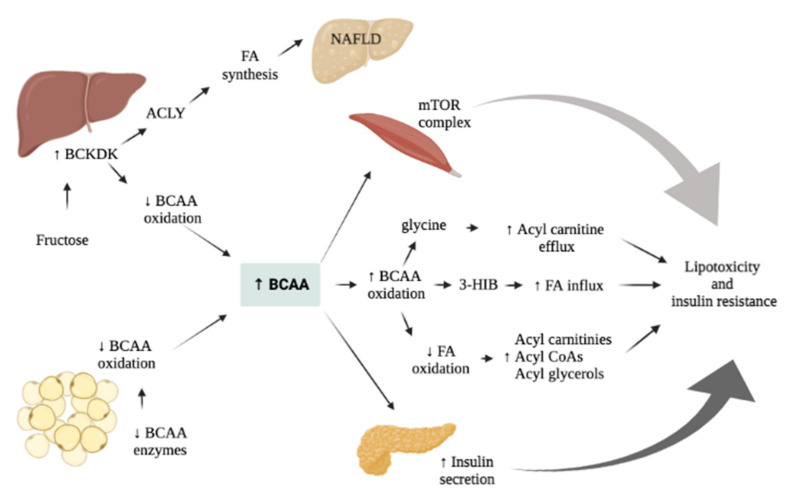
Main mechanisms proposed linking BCAAs and metabolic alterations associated to obesity in children and adolescents. Fructose stimulates the activation of BCKDK in the liver, which enhances lipogenesis and reduces BCAAs oxidation. This leads to an increase of BCAAs’ levels, which may facilitate (a) mTOR complex activation in the muscle, (b) insulin secretion and (c) the production of metabolites. These effects finally result in lipotoxicity and insulin resistance. Abbreviations. ACLY: ATP Citrate Lyase; BCAAs: Branched Chain Amino Acids; BCKDK: Branched Chain Ketoacid Dehydrogenase Kinase; FA: Fatty Acids; mTOR: mammalian target of rapamycin; 3-HIB: 3-Hydroxyisobuterate.

## Data Availability

Not applicable.

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
