# Peer review of "Amino Acid-Related Metabolic Signature in Obese Children and Adolescents"

_nutrients, 2022, doi:10.3390/nu14071454_

Round 1

Reviewer 1 Report

In the present review, Nella Polidori and co-workers performed a revision of the literature to assess the role of branched-chain amino acids (BCAAs) in children and adolescents’ metabolism, focusing on the molecular pathways involved. The authors concluded that these compounds may help in the precocious identification of many complications of pediatric obesity. However, further studies are still needed to better assess if BCAAs may be used to screen these conditions and if any other metabolomic compound may be useful to achieve this goal. Overall, I think that the manuscript is well-structured (within the scope of "Nutrients”) and of clinical impact on a current topic of interest.

I have a small suggestion/curiosity to improve the quality of review.

In light of the current evidence, please clarify if the BCAAs levels correlate with different dietary habits in children and adolescents. In other words, it is possible that these compounds may help to identify the possible positive/negative effects of a "diet"? Please discuss this intriguing topic.

Author Response

We thank the Reviewer for his/her comments, we fully agree with his/her view and with the novelty added to the review by evaluating this issue. Therefore, we have added a paragraph about the main points characterizing the relationship between diet and BCAAs levels, focusing on its suitable role in monitoring the quality of the diet followed by the patients (lines 497-516)

Reviewer 2 Report

Thank you very much for this interesting overview on AA signature in obesity. As metabolomics capabilities continue to improve, this work provides a good overview.

I have the following comments:

  1. Please check the citations in the text e.g. et al. the point is often missing.
  2. Please check the grammar and text style e.g. line 69 to 71.
  3. Line 80 to 81: It should be mentioned that AA measurements are quite expensive in some countries and are not part of the normal services provided in a clinic.  Some metabolomic analyses are not daily routine. The comparison of liver fat content results is only valid using the same protocol and the same devices to assess liver fat content as well as cohort/ethnic comparison.
  4. Line 94 to 95: What are the differences for the authors between racial and ethnic factors?
  5. Line 103: Same as above
  6. Line 128: In fact,...
  7. Line 150: Check grammar style.
  8. Line 195: Please include an explanation why there are sex differences.
  9. Line 278: A figure would aid in understanding the text.
  10. Line 327: In detail,...
  11. Figure 1: Please include an explanation of the sympoles and pathways. Furthermore, abbreviations should be named.
  12. Same for Figure 2.
  13. Line 430: Check alfa and beta-hydroxy-430 butirrate.

Author Response

Thank you very much for this interesting overview on AA signature in obesity. As metabolomics capabilities continue to improve, this work provides a good overview.

I have the following comments:

ï‚·  Please check the citations in the text e.g. et al. the point is often missing.

We thank the Reviewer for his/her suggestion. We have modified the text adding the point when missing.

ï‚·  Please check the grammar and text style e.g. line 69 to 71.

We thank the Reviewer for his/her suggestion. We have checked the grammar and the text style, modifying it accordingly.

ï‚·  Line 80 to 81: It should be mentioned that AA measurements are quite expensive in some countries and are not part of the normal services provided in a clinic. Some metabolomic analyses are not daily routine. The comparison of liver fat content results is only valid using the same protocol and the same devices to assess liver fat content as well as cohort/ethnic comparison.

 We thank the Reviewer for his/her correct suggestion and we completely agree with his/her statement. We have inserted a comment about it (lines 82-85).

ï‚·  Line 94 to 95: What are the differences for the authors between racial and ethnic factors?

We thank the Reviewer for his/her suggestion. We have used always the term ethnic through the manuscript in order to avoid misinterpretation.

ï‚·  Line 103: Same as above

We thank the Reviewer for his/her suggestion. We have used always the term ethnic through the manuscript in order to avoid misinterpretation.

ï‚·  Line 128: In fact,

We apologize for the typing error, we have corrected the mistake.

ï‚·  Line 150: Check grammar style.

We apologize for the grammar error; we have corrected the mistake.

ï‚·  Line 195: Please include an explanation why there are sex differences.

We thank the Reviewer for his/her suggestion. We have added an explanation about the higher preponderance of NAFLD in males (lines 201-205)

ï‚·  Line 278: A figure would aid in understanding the text.

            We thank the Reviewer for his/her suggestion. We have added a new figure (Figure 1).

ï‚·  Line 327: In detail...

We have modified the text as suggested (line 339).

ï‚·  Figure 1: Please include an explanation of the symbols and pathways. Furthermore, abbreviations should be named.

          We thank the Reviewer for his/her suggestion. We have added an explanation on the figure (now figure 2).

ï‚·  Same for Figure 2.

We thank the Reviewer for his/her suggestion we have provided an explanation for the figure (now figure 3).

ï‚·  Line 430: Check alfa and beta-hydroxy-430 butirrate.

We apologize for the grammar mistake; we have corrected the mistake (line 454).

Round 2

Reviewer 2 Report

Thank you very much for the edited version of the manuscript.

I have just one comment; instead of figure 1 showing the AAs a representation of lines 292 to 300 would be preferable (like figure 2).